# Generalized Bayesian Quadrature with Spectral Kernels

**Houston Warren**[1]  **Rafael Oliveira**[2]  **Fabio Ramos**[1, 3]

[1]School of Computer Science, The University of Sydney, Sydney, Australia
[2]Brain and Mind Centre, The University of Sydney, Sydney, Australia
[3]NVIDIA, USA

## Abstract

Bayesian probabilistic integration, or Bayesian quadrature (BQ), has arisen as a popular means of numerical integral estimation with quantified uncertainty for problems where computational cost limits data availability. BQ leverages flexible Gaussian processes (GPs) to model an integrand which can be subsequently analytically integrated through properties of Gaussian distributions. However, BQ is inherently limited by the fact that the method relies on the use of a strict set of kernels for use in the GP model of the integrand, reducing the flexibility of the method in modeling varied integrand types. In this paper, we present spectral Bayesian quadrature, a form of Bayesian quadrature that allows for the use of *any* shift-invariant kernel in the integrand GP model while still maintaining the analytical tractability of the integral posterior, increasing the flexibility of BQ methods to address varied problem settings. Additionally our method enables integration with respect to a uniform expectation, effectively computing definite integrals of challenging integrands. We derive the theory and error bounds for this model, as well as demonstrate GBQ's improved accuracy, flexibility, and data efficiency, compared to traditional BQ and other numerical integration methods, on a variety of quadrature problems.

## 1 INTRODUCTION

Methods for estimation of non-analytical integrals through numerical methods play a key role across a broad spectrum of scientific fields, but these methods are often computationally expensive in nature. Methods such as finite-element or volumes, which are widely used in physical simulation to integrate partial differential equations, or Monte Carlo estimation, which is widely used in Bayesian statistics for estimation of posteriors, require a large number of function evaluations to reach a desired level of accuracy. In addition, many numerical integration methods fail to provide uncertainty quantification on their estimates, which is crucial in the applied settings in which physical simulation is often used.

Bayesian quadrature (BQ) (Diaconis, 1988; O'Hagan, 1991) is a probabilistic method which can remedy these concerns by offering performance on computationally-limited small data while admitting robust uncertainty bounds. BQ takes the form of a traditional quadrature rule:

$$\int f(\boldsymbol{x})p(\boldsymbol{x})d\boldsymbol{x} \approx \sum_{i=1}^{n} w_i f(\boldsymbol{x}_i), \qquad (1)$$

for $n$ evaluations of the function $f$, where weights $w_i \sim p(\boldsymbol{x})$ are instead learned through manipulation of a Bayesian non-parametric Gaussian process (GP) (C. E. Rasmussen and Williams, 2006) model on observations of the integrand $f(\boldsymbol{x})$.

The use of such a Bayesian non-parametric model for learning weights leverages the ability for GPs to perform well under data-scarcity as well as quantify uncertainty in a principled manner. In addition, the Gaussian nature of this model allows for the integral estimate of $f$ to be a simple analytical integration of the GP prior on $f$ using well-known characteristics of multi-variate Gaussian distributions. Previous work (Ghahramani and C. Rasmussen, 2003; Kandasamy, Schneider, and Póczos, 2015) has clearly demonstrated computational efficiency of BQ versus traditional methods such as Monte Carlo integration when the data dimensionality $d < 10$.

A chief advantage of using GPs in any probabilistic learning setting is the flexibility of choice of the GP kernel function $k$, which allows for a practitioner to inject domain knowledge of the problem space into the GP model. Characteristics such as as data smoothness or periodicity can easily be applied through choice or composition of specific kernel

*Accepted for the 38th Conference on Uncertainty in Artificial Intelligence* (UAI 2022).

functions tailored to these settings.

However, the traditional BQ formulation hamstrings this flexibility by limiting the choice of kernel in the integrand GP to only a small subset of kernels with known analytical kernel means, such as Gaussian or polynomial kernels. For well-known kernels that may not be analytically tractable in the BQ setting, but nonetheless might better model an integrand, traditional numerical quadrature methods must be used, reducing the computational efficiency that BQ offers. The question naturally arises of how practitioners might enable the full suite of kernel choices for use in the GP integrand prior while still maintaining the analytical tractability in the BQ setting, to most efficiently produce an accurate estimate to the integral of $f$.

In this paper, we expand on the literature of BQ and propose a solution to the problem of kernel choice with generalized Bayesian quadrature (GBQ), a method derived from random Fourier features (RFFs) by which any shift-invariant kernel can be used in the GP integrand prior while still allowing for analytical tractability in the BQ setting. By allowing for both kernel flexibility and analytical integration, we expand upon the ability of traditional BQ to model a variety of integrand types while still maintaining the computational efficiency BQ offers. We summarize our contributions here:

**Contributions**

- We propose generalized Bayesian quadrature (GBQ), a method of Bayesian quadrature that allows for the use of *any* shift-invariant kernel in the GP model of the integrand while still admitting an analytical estimate of the integral posterior mean and variance.

- We show that GBQ can directly be used to compute integrals over Gaussian and uniform measures within the same framework.

- We derive the upper-bounded error to this approximation as a function of data-availability.

- We outline the assumptions under which GBQ shares the computational complexity of traditional BQ.

- We demonstrate the accuracy and flexibility of this quadrature method versus traditional BQ, as well as data-efficiency versus typical Monte Carlo integration, on a selection of relevant domain problems.

## 2 RELATED WORKS

Quadrature methods of the type in equation 1 are well-studied due to their importance to a variety of fields, and there is a deep literature dating back centuries on methods for numerically approximating integrals. We will briefly review here relevant methods in relation to Bayesian quadrature.

Rather than deterministic quadrature weighting, various probabilistic quadrature approaches have been proposed (C. J. Oates and Sullivan, 2019) for integration when model observations are expensive, with one of the most popular methods being Bayesian quadrature. Many extensions to vanilla BQ have been developed over the years to improve performance and provide theoretical guarantees (Acerbi, 2018; Belhadji, Bardenet, and Chainais, 2019; Briol, Chris J. Oates, Girolami, and M. A. Osborne, 2015; Kennedy, 1998). Other applications include use in multi-fidelity modeling (Gessner, Gonzalez, and Mahsereci, 2020), Bayesian posterior estimation (Gunter et al., 2014; M. Osborne, Garnett, Roberts, et al., 2012), Bayesian optimization (Nguyen et al., 2020), and model selection (Chai et al., 2019; M. Osborne, Garnett, Ghahramani, et al., 2012).

The derivation of analytical forms, or empirical approximation, of kernel means, which is a significant component of the BQ formulation, is a problem that appears in numerous other fields. Namely, kernel mean embedding (Muandet et al., 2017), deep Gaussian processes (Damianou and Lawrence, 2013), and neural operators (Kovachki et al., 2021; Li et al., 2021) all attempt to do so through various means. There also exist empirical methods for the estimation of kernel means using random Fourier features (Muandet et al., 2017), as well as strong theoretical connections between the very concept of kernel-based quadrature and random Fourier features (Bach, 2017). In a related manner, methods have been proposed that seek to implement Fourier feature kernels through quadrature based methods (Mutny and Krause, 2018). However, to our knowledge, no methods directly solve kernel integrals analytically in the BQ setting using RFFs, as we propose to do in this paper.

Similar to RFFs, spectral mixture kernels (SMKs) (Oliva et al., 2016; Wilson and Adams, 2013), also model shift-invariant kernels as the spectral transform of a probability measure. In the SMK case, this measure is a Gaussian mixture model, which can be shown to asymptotically approximate any stationary kernel as the number of mixture components increases. While BQ has found applications in constructing hyper-kernels by marginalizing SMKs over mixture priors (Hamid et al., 2022), their use within BQ has been limited.

The method which shares the most overlap with this work is the Fourier neural operator (FNO) (Li et al., 2021), which, as a part of a larger deep neural network architecture, estimates the convolution of shift-invariant kernels with a probability measure using parameters in Fourier space. While we take a similar approach to deriving kernel means using Fourier frequencies, the overall frameworks differ, with GBQ existing in the Gaussian process framework, thus offering uncertainty estimates for integral posteriors, while FNOs exist within a deterministic neural network architecture.

# 3 PRELIMINARIES

## 3.1 BAYESIAN QUADRATURE

We will now review various preliminary methods upon which GBQ is built, starting with Bayesian quadrature.

BQ assumes we have a function $f$ that we are trying to integrate and a dataset $\mathcal{D} = \{x_i, y_i\}_{i=1}^n$ with $n$ noisy observations of $f$, where $x \in \mathcal{R}^d$, $y_i = f(x_i) + \epsilon$, and $\epsilon$ is i.i.d normal distributed noise. Typically, $f$ is computationally expensive to evaluate, implying a small $n$ and highlighting the need for uncertainty estimation in the final integral approximation. BQ does this by first placing a Gaussian process (C. E. Rasmussen and Williams, 2006) prior on $f$, which we will briefly review here.

**Gaussian Processes** Gaussian processes are a Bayesian non-parametric method which model the target data generation function $f$ we are attempting to learn as a joint multivariate Gaussian of the form:

$$f \sim \mathcal{GP}(\boldsymbol{\mu}(x), k_{\boldsymbol{\theta}}(x, x')), \qquad (2)$$
$$y = f(x) + \epsilon, \qquad (3)$$

where $k_{\boldsymbol{\theta}}$ is a positive semi-definite *kernel function* with hyper-parameters $\boldsymbol{\theta}$, and $\boldsymbol{\mu}$ is a mean function. In the above, we assume an additive and independent Gaussian noise observation model with, $\epsilon \sim \mathcal{N}(0, \sigma^2 I)$, where $y$ are noisy observations with standard deviation $\sigma$. $k$ is typically chosen *a-priori* to encode known characteristics of the data $\mathcal{D}$ such as periodicity and smoothness.

For inference, the posterior-predictive distribution of $f_*$ for a new data point $\{x_*\}$, given the training data $\mathcal{D} = \{x^i, y^i\}_{i=1}^n$, and Gram matrix $K_{xx} = k_{\boldsymbol{\theta}}(x, x'), \forall x, x'$, is given by $\mathcal{N}(\boldsymbol{\mu}(f*), \text{Cov}(f*))$ where,

$$\mu(f*) = K_{*x}(K_{xx} + \sigma^2 I)^{-1} y, \quad (4)$$
$$\text{Cov}(f*) = K_{**} - K_{*x}(K_{xx} + \sigma^2 I)^{-1} K_{x*}. \quad (5)$$

In BQ, by setting a GP prior on the integrand $f$ we can leverage the ability of GPs flexibly and accurately model functions with uncertainty on small data, but it is also advantageous in that we can directly and analytically integrate the integrand GP prior. This is performed using well-known characteristics of Gaussian distributions in order to form a posterior estimate $\langle \bar{f} \rangle$ of the integral of $f$.

Formally, the mean of the BQ estimate of $\langle \bar{f} \rangle$ is the expected value over measure $p(x)$ of the posterior mean of the GP prior (4) on $f$:

$$
\begin{aligned}
\langle \bar{f} \rangle &= \int_{x \in \mathcal{R}} k(x, X)^T K^{-1} y \, p(x) \, dx \\
&= y^T K^{-1} \int_{x \in \mathcal{R}} k(x, X) \, p(x) \, dx \qquad (6) \\
&= \mu_x(X)^T K^{-1} y,
\end{aligned}
$$

where $\mu_x(X) = [\mu_x(x_1) \dots \mu_x(x_n)]$ can be seen as the *kernel mean* over measure $p(x)$. The variance of this estimate is:

$$\mathbb{V}(\langle \bar{f} \rangle) = \int_{X \in \mathcal{R}^d} \mu_x(X) p(X) \, dX \qquad (7)$$

which is notably independent of prior observations $X$.

The mean formulation mirrors that of standard quadrature methods shown in equation (1), differing in that weights $\mu_x(X)^T K^{-1}$ are the result of probabilistic learning on observed data $\mathcal{D}$ and associated kernel choice, rather than decided *a priori* or by a heuristic.

Under a very limited selection of kernel and sampling measure choices, the mean (6) and variance (7) can be calculated analytically (Briol, Chris J. Oates, Girolami, M. A. Osborne, and Sejdinovic, 2019). Most commonly, a Gaussian kernel and Gaussian distribution for the measure $p(x)$, as proposed by (O'Hagan, 1991), is one such case. It is also prudent to note that the measure distribution can be fluid while retaining analytical tractability through use of importance sampling (Briol, Chris J. Oates, Cockayne, et al., 2017; Ghahramani and C. Rasmussen, 2003), while the choice of kernel is more restricted.

In BQ, the limitation of the kernel to certain forms dependent on known closed-form analytical integration over the measure $p(x)$ gives up one of the greatest advantages of the GP prior: flexible selection of kernels for specific domains. To alleviate this issue, GBQ introduces random Fourier features into the BQ formulation for parametrization of the GP kernel.

## 3.2 RANDOM FOURIER FEATURES

As we shall see in Section 4 Random Fourier features enable the use of *any* shift-invariant kernel in the BQ-GP prior without sacrificing the analytical tractability of the integral posterior. This greatly increases the flexibility of the BQ to perform under a variety of problem conditions for which different kernels may be necessary.

Random Fourier features are obtained from the spectral representation of shift-invariant kernels given by Bochner's theorem:

**Theorem 1** (Bochner's theorem (Rudin, 2011)). *A shift-invariant kernel $k(x, x') = k(x - x')$ is positive-definite if and only if it is the Fourier transform of a non-negative measure.*

Theorem 1 is the building block upon which (Rahimi and Recht, 2008) introduce random Fourier features (RFFs), which define a practical means by which Bochner's theorem can be applied in practice to estimate kernel functions in finite dimensions. Using the derivation from (Rahimi and

Recht, 2008), if the probability density $p(\boldsymbol{\omega})$ is the Fourier transform of $k$:

$$
\begin{aligned}
k(\boldsymbol{x} - \boldsymbol{x}') &= \int_{\mathcal{R}^d} p(\boldsymbol{\omega}) e^{j\boldsymbol{\omega}(\boldsymbol{x} - \boldsymbol{x}')}\, d\boldsymbol{\omega}, \\
&= \int_{\mathcal{R}^d} p(\boldsymbol{\omega}) \cos(\boldsymbol{\omega}(\boldsymbol{x} - \boldsymbol{x}'))\, d\boldsymbol{\omega}.
\end{aligned}
\tag{8}
$$

For brevity, equation (8) provides the formulation for the case that the kernel and data $\boldsymbol{x}$ are real-valued, but an alternative formulation exists for the case they are not.

It can be easily seen that the kernel function $k$ is entirely defined by the choice of density $p(\boldsymbol{\omega})$, and several common kernels have known associated densities. For example, if $p(\boldsymbol{\omega})$ is multivariate isotropic Gaussian, then (8) represents the radial basis function (RBF) kernel. By drawing from the associated $p(\boldsymbol{\omega})$ for our choice of kernel, RFFs approximate (8) with Monte Carlo by:

$$
k(\boldsymbol{x}, \boldsymbol{x}') = k(\boldsymbol{x} - \boldsymbol{x}') \approx \frac{1}{R} \sum_{r=1}^{R} \cos(\boldsymbol{\omega}_r^T (\boldsymbol{x} - \boldsymbol{x}')) \quad (9)
$$

where $R$ is the number of Monte Carlo samples or *Fourier features*.

Alternatively, we can directly parametrize these features $\boldsymbol{\omega}$ as GP hyperparameters, which allows for optimal kernels to learned during training to best adapt to specific problem settings (Chang et al., 2017; Oliva et al., 2016; Tompkins et al., 2019; Zhen et al., 2020).

# 4 GENERALIZED BAYESIAN QUADRATURE

We build upon these concepts to devise our method, generalized Bayesian quadrature, which enables flexible Bayesian quadrature for use with any arbitrary shift-invariant kernel while maintaining analytical tractability of the kernel mean $\mu_{\boldsymbol{x}}(\boldsymbol{X})$. We begin by showing that a Gaussian density can be approximated with RFFs, which will lead to analytical tractability for general shift-invariant kernels.

## 4.1 PROBABILITY DENSITY FUNCTIONS AS RFF KERNELS

Analytical tractability of the BQ mean in (6) for any kernel represented by RFFs can be achieved by reformulating the kernel mean measure $p(\boldsymbol{x})$ as an RFF as well. In general, we can turn any positive-definite probability density function $p : \mathcal{X} \to [0, \infty)$ on $\mathcal{X} \subseteq \mathbb{R}^d$ into a stationary kernel via the following construction:

$$
\begin{aligned}
k_p &: \mathcal{X} \times \mathcal{X} \longrightarrow \mathbb{R}, \\
k_p(\boldsymbol{x}, \boldsymbol{x}') &\longmapsto \begin{cases} p(\boldsymbol{x} - \boldsymbol{x}'), & \boldsymbol{x} - \boldsymbol{x}' \in \mathcal{X}, \\ 0, & \boldsymbol{x} - \boldsymbol{x}' \notin \mathcal{X}. \end{cases}
\end{aligned}
\tag{10}
$$

It is easy to verify that a kernel defined as in equation 10 is translation-invariant and positive-definite whenever $p$ is. As examples of distributions with positive-definite densities we have the Gaussian and the Student-T (Rossberg, 1995). Given that many probability distributions can be approximated by these densities, or mixtures of them, kernel modeling of distributions as in (10) has a wide range of potential applicability.

**RFF Representation of the Gaussian**  Given that an RBF kernel represents an un-normalized Gaussian, by sampling $\boldsymbol{\rho}$ from $\mathcal{N}(\boldsymbol{0}, \boldsymbol{I})$ and using a multivariate Gaussian normalizing constant $\tau^{-1} = [(2\pi)^d |\Sigma|]^{-1/2}$, where $\Sigma$ is the length-scale matrix for features $\boldsymbol{\rho}$, we can formulate an RFF kernel approximation of a Gaussian density function $q(\boldsymbol{x})$ as $\lim_{R \to \infty}$ as:

$$
\begin{aligned}
p(\boldsymbol{x}) \approx q(\boldsymbol{x}) &= \tau^{-1} \exp\{-|\boldsymbol{x} - \boldsymbol{\mu}|^2\} \\
&\approx [(2\pi)^d |\boldsymbol{\Sigma}|]^{-1/2} \frac{1}{R} \sum_{r=1}^{R} \cos(\boldsymbol{\rho}_r^T (\boldsymbol{x} - \boldsymbol{\mu})).
\end{aligned}
\tag{11}
$$

This form allows for the use of simple trigonometric identities to form an analytically integrable kernel mean formulation (6) over a Gaussian measure, which we will shortly demonstrate.

## 4.2 GENERALIZED BAYESIAN QUADRATURE POSTERIOR

We now reformulate the BQ mean and variance by substituting the RFF formulations of both the kernel and measure in equations (9) and (11) into the BQ mean in equation (6).

$$
\begin{aligned}
\langle \bar{f} \rangle = \boldsymbol{y}^T \boldsymbol{K}^{-1} &\int_{\boldsymbol{x} \in \mathcal{R}} \frac{1}{R} \sum_{r=1}^{R} \cos(\boldsymbol{\omega}_r^T (\boldsymbol{x} - \boldsymbol{X})) \\
&\times [(2\pi)^d |\boldsymbol{\Sigma}|]^{-1/2} \frac{1}{Z} \sum_{z=1}^{Z} \cos(\boldsymbol{\rho}_z^T (\boldsymbol{x} - \boldsymbol{\mu})) d\boldsymbol{x} \quad (12)
\end{aligned}
$$

The trigonometric form of both the kernel and measure distribution in this setting allow for the application of basic identities to rewrite the integrand as a linear function. Using the identity $\cos(\alpha)\cos(\beta) = \cos(\alpha + \beta)/2 + \cos(\alpha - \beta)/2$, and simple properties regarding the anti-derivatives of trigonometric functions, we arrive at the following definition of GBQ over an approximated Gaussian measure $q(\boldsymbol{x})$.

**Definition 1** (Generalized Bayesian Quadrature Over Gaussian Measures). *Given $n$ noisy observations $\{\boldsymbol{x}_i, y_i\}_{i=1}^{n} = \{\boldsymbol{X}, \boldsymbol{y}\}$ of a function $f$ where $\boldsymbol{x}_i \in \mathcal{R}^d$, a kernel function $k$ parametrized through random Fourier frequencies $\boldsymbol{\omega} \in \mathcal{R}^{R \times d}$ sampled from density $p(\boldsymbol{\omega})$, a Gaussian measure approximation $q(\boldsymbol{x})$ parametrized by Fourier frequencies $\boldsymbol{\rho} \in \mathcal{R}^{Z \times d}$ sampled from $\mathcal{N}(\boldsymbol{0}, \boldsymbol{I})$, and kernel matrix*

$\boldsymbol{K} = [k(\boldsymbol{x}_i, \boldsymbol{x}'_j)]^n_{i,j=1} \in \mathbb{R}^{n \times n}$, the GBQ estimate $\langle \bar{f} \rangle$ of the mean of the integral of $f$ over domain $\boldsymbol{a} \leq \boldsymbol{x} \leq \boldsymbol{b}$ is:

$$\langle \bar{f} \rangle = \mu_{\boldsymbol{x}}(\boldsymbol{X})^T \boldsymbol{K}^{-1} \boldsymbol{y}, \qquad (13)$$

$$\mu_{\boldsymbol{x}}(\boldsymbol{X}) =$$
$$L \sum_{r=1}^{R} \sum_{z=1}^{Z} \frac{h^d(\boldsymbol{x}^T(\boldsymbol{\omega}_r + \boldsymbol{\rho}_z) - (\boldsymbol{\omega}_r^T \boldsymbol{X} + \boldsymbol{\rho}_z^T \boldsymbol{\mu}))}{\prod_{j=1}^{d}(\omega_r^j + \rho_z^j)} \Bigg|_{\boldsymbol{a}}^{\boldsymbol{b}} +$$
$$L \sum_{r=1}^{R} \sum_{z=1}^{Z} \frac{h^d(\boldsymbol{x}^T(\boldsymbol{\omega}_r - \boldsymbol{\rho}_z) - (\boldsymbol{\omega}_r^T \boldsymbol{X} - \boldsymbol{\rho}_z^T \boldsymbol{\mu}))}{\prod_{j=1}^{d}(\omega_r^j - \rho_z^j)} \Bigg|_{\boldsymbol{a}}^{\boldsymbol{b}},$$
$$(14)$$

where $d$ is the dimensionality of $\boldsymbol{x}$, and $h^d$ is the function at the $d$-th index of the repeating series $h = [\sin, -\cos, -\sin, \cos, \sin, \dots]$. The normalization constant $L$ is defined as:

$$L = (2RZ \times Q_{\boldsymbol{a}}^{\boldsymbol{b}})^{-1}[(2\pi)^d |\boldsymbol{\Sigma}|]^{-1/2}, \qquad (15)$$

where $Q_{\boldsymbol{a}}^{\boldsymbol{b}} = \int_{\boldsymbol{a}}^{\boldsymbol{b}} q(\boldsymbol{x}) d\boldsymbol{x}$ is an estimate to the CDF of the RFF-parametrized Gaussian $q(\boldsymbol{x})$, which is analytically calculable from (11) [1].

See the supplement for full proof, variance derivation, and details of an algorithm for efficient implementation. In addition, the supplement provides a GBQ formulation over uniform measures, which equates to direct integration of the GP integrand $\bar{f}$. Through definition 1 we obtain an analytical posterior for $\langle \bar{f} \rangle$ and $\mathbb{V}(\langle \bar{f} \rangle)$ that allows for flexible kernel choice through the use of RFFs.

### 4.3 APPROXIMATION ERROR

#### 4.3.1 Gaussian Process and Random Fourier Features Error Bounds

The approximation error of GBQ extends from well-known error bounds derived from the literature of RFFs and BQ respectively. We present here an abbreviated form of this proof, the full version of which can be found in the supplement.

We begin with the following lemma outlining the error of the GP estimate $\bar{f}$ to the integrand $f$ under the assumption $f$ is a member of the Hilbert space $\mathcal{H}_k$ defined by kernel $k$:

**Lemma 1** (Durand, Maillard, and Pineau (2017, Theorem 1)). *Assume $f \in \mathcal{H}_k$ and that the observation noise $\epsilon$ is $\sigma_\epsilon$-sub-Gaussian. Then the following holds with probability at least $1 - \delta$:*

$$\forall n \in \mathbb{N}, |f(\boldsymbol{x}) - \mu_n(\boldsymbol{x})| \leq \beta_k(\delta)\sigma_n(\boldsymbol{x}), \forall \boldsymbol{x} \in \mathcal{X}, \quad (16)$$

---

[1]See supplementary for derivation.

where $\mu_n$ and $\sigma_n^2$ denote the GP posterior mean and variance given $n$ observations, according to (4) and 5, respectively, and

$$\beta_k(\delta) := \|f\|_k$$
$$+ \sigma_\epsilon \sqrt{\frac{2}{\lambda} \log \left( \frac{\det(\mathbf{I} + \lambda^{-1}\mathbf{K}_n)^{1/2}}{\delta} \right)}, \quad (17)$$

with

$$\mathbf{K}_n := [k(\boldsymbol{x}_i, \boldsymbol{x}'_j)]^n_{i,j=1} \in \mathbb{R}^{n \times n}, \qquad (18)$$

We follow with a lemma related to the error bounds on the RFF approximation to a shift-invariant kernel $k$.

**Lemma 2** (Sutherland and Schneider (2015, Proposition 1)). *Let $k : \mathcal{X} \times \mathcal{X} \to \mathbb{R}$ be a continuous shift-invariant positive-definite kernel with $k(\boldsymbol{x}, \boldsymbol{x}) = 1$ and such that $\nabla^2 k(\boldsymbol{x}, \boldsymbol{x})$ exists, for all $\boldsymbol{x} \in \mathcal{X} \subset \mathbb{R}^d$. Suppose $\mathcal{X}$ is compact with diameter $\ell_{\mathcal{X}} < \infty$. Denote $k$'s Fourier transform as $P_k$, which is a probability measure, and let $\sigma_k^2 := \mathbb{E}[\|\boldsymbol{\omega}\|_2^2]$ for $\omega \sim P_k$. Let $\tilde{k} : \mathcal{X} \times \mathcal{X} \to \mathbb{R}$ denote $k$'s RFF approximation with $R$ frequencies according to (9). Then the following holds for any $0 < \xi < \sigma_k \ell_{\mathcal{X}}$:*

$$\mathbb{P}\left[ \sup_{\boldsymbol{x}, \boldsymbol{x}' \in \mathcal{X}} |\tilde{k}(\boldsymbol{x}, \boldsymbol{x}') - k(\boldsymbol{x}, \boldsymbol{x}')| \geq \xi \right]$$
$$\leq 66 \left( \frac{\sigma_k \ell_{\mathcal{X}}}{\xi} \right)^2 \exp \left( -\frac{R\xi^2}{4(d+2)} \right). \quad (19)$$

*Therefore, for any $\delta \in (0, 1)$, we can achieve pointwise approximation error less than $\xi$ with probability at least $1 - \delta$ if:*

$$R \geq R(\xi, \delta, \sigma_k) := \frac{4(d+2)}{\xi^2} \left( \frac{2}{1 + \frac{2}{d}} \log \frac{\sigma_k \ell_{\mathcal{X}}}{\xi} + \log \frac{66}{\delta} \right) \quad (20)$$

#### 4.3.2 Generalized Bayesian Quadrature Error

Next, we formulate an error bound on the RFF parametrization of the Gaussian (or any arbitrary) density shown in equation (11), as we build towards a final bound on GBQ.

**Theorem 2** (Error of the RFF Density Approximation). *Let $p : \mathcal{X} \to \mathbb{R}$ be a positive-definite probability density function defined on $\mathcal{X} \subset \mathbb{R}^d$ which is such that $\nabla^2 p(\mathbf{0})$ exists. Assume $\mathcal{X}$ is compact, and let $b_p > 0$ be any constant such that $b_p \geq \max_{\boldsymbol{x} \in \mathcal{X}} p(\boldsymbol{x})$. Let $\tilde{k}_p$ denote an RFF approximation with $Z \in \mathbb{N}$ frequencies to $k_p$ as defined in (10), and let $\tilde{p} : \boldsymbol{x} \mapsto \tilde{k}_p(\boldsymbol{x}, \mathbf{0})$, $\boldsymbol{x} \in \mathcal{X}$. Then, for any $\xi > 0$, the following holds:*

$$\mathbb{P}\left[ \sup_{\boldsymbol{x} \in \mathcal{X}} |\tilde{p}(\boldsymbol{x}) - p(\boldsymbol{x})| \geq b_p \xi \right]$$
$$\leq 66 \left( \frac{\sigma_{k_p} \ell_{\mathcal{X}}}{\xi} \right)^2 \exp \left( -\frac{Z_p \xi^2}{4(d+2)} \right) \quad (21)$$

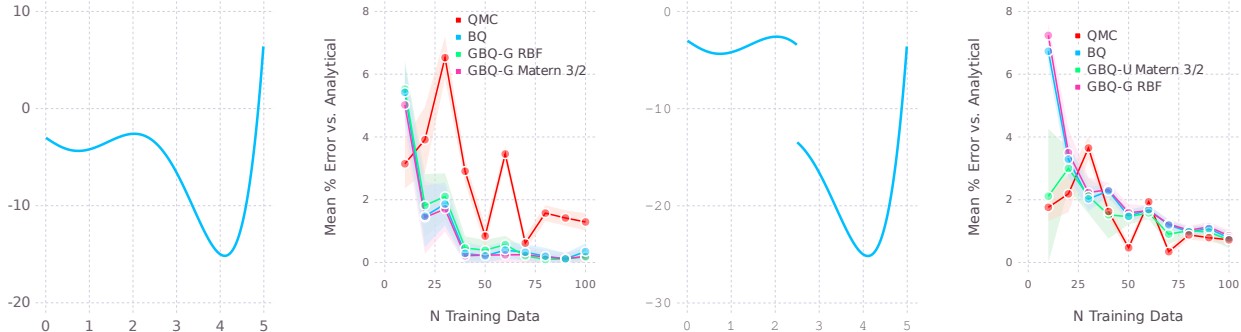

Figure 1: Function Plots and Bounded Error Graphs for 1D Continuous and Disjoint Polynomial Quadrature Experiments.

*where for the second statement we assume $\xi \leq \sigma_{k_p} \ell_{\mathcal{X}}$, and $\sigma_{k_p}$, $\ell_{\mathcal{X}}$, $\alpha_\xi$ and $\beta_\xi$ are the same as defined in [Lemma 2](#) for $k := \frac{1}{b_p} k_p$.*

Finally, we combine these results to arrive at the upper bounded error for GBQ as a composition of the errors of GP approximation, RFF approximation, and RFF measure density estimation.

**Theorem 3** (Upper-Bounded Generalized Bayesian Quadrature Error). *Let $f \in \mathcal{H}_k$, where $k : \mathcal{X} \times \mathcal{X} \to \mathbb{R}$ is a positive-definite, translation-invariant kernel on $\mathcal{X} \subset \mathbb{R}^d$. Assume that:*

1. *$\mathcal{X}$ is compact with diameter $\ell_{\mathcal{X}} < \infty$ and volume $v_{\mathcal{X}} := \int_{\mathcal{X}} \mathrm{d}\boldsymbol{x} < \infty$;*
2. *$k(\boldsymbol{0}, \boldsymbol{0}) = 1$ and $\nabla^2 k(\boldsymbol{0}, \boldsymbol{0})$ exists;*
3. *and $p : \mathcal{X} \to [0, \infty)$ is a positive-definite probability density function.*

*Then the following holds with probability at least $1 - \delta$:*

$$
\left| \int_{\mathcal{X}} f(\boldsymbol{x}) p(\boldsymbol{x}) \, \mathrm{d}\boldsymbol{x} - \int_{\mathcal{X}} \tilde{\mu}_n(\boldsymbol{x}) \tilde{p}(\boldsymbol{x}) \, \mathrm{d}\boldsymbol{x} \right|
$$
$$
\leq \left( \frac{n}{\lambda} \beta_\epsilon \left( \frac{\delta}{4} \right) \xi_k + \beta_k \left( \frac{\delta}{4} \right) \max_{\boldsymbol{x} \in \mathcal{X}} \sigma_n(\boldsymbol{x}) \right)
$$
$$
\times (1 + b_p \xi_p v_{\mathcal{X}}) + \|f\|_\infty b_p \xi_p v_{\mathcal{X}}, \quad (22)
$$

*where $\beta_\epsilon(\delta) := \|f\|_\infty + \sigma_\epsilon \sqrt{2 \log \left( \frac{n}{\delta} \right)}$, for an RFF approximation to $k$ with $R \geq R\left( \xi_k, \frac{\delta}{4}, \sigma_k \right)$ frequencies and an RFF approximation to $p$ with $Z \geq R\left( \xi_p, \frac{\delta}{4}, \sigma_{k_p} \right)$ frequencies, given $\xi_k > 0$ and $\xi_p > 0$.*

We refer the reader to the supplementary for the full proof of theorems 2 and 3.

### 4.4 COMPLEXITY

We consider here the complexity of calculating the BQ and GBQ mean integral approximation $\langle \bar{f} \rangle$ as in 6 and 14.

Traditional BQ over a Gaussian measure $p(\boldsymbol{x})$ (6), under the assumption $N > d$, has a mean-calculation complexity that is dominated by the operation $\boldsymbol{K}^{-1}$, which scales in $\mathcal{O}(N^3)$.

Comparatively, GBQ can be either dominated by the same term or via the complexity introduced through the novel method of estimation of the RFF kernel mean as in definition 1.

GBQ mean calculation with a Gaussian measure, over definite bounds of dimensionality $d$ for all $N$, scales in $\mathcal{O}(dNRZ)$ with the number of Fourier features $R$ used for kernel $k$ approximation, and the number of Fourier features $Z$ used for RFF approximation $q(\boldsymbol{x})$ of Gaussian measure $p(\boldsymbol{x})$. Over a uniform measure and definite bounds, which equates to direct integration of the integrand GP $\bar{f}$, calculation of the GBQ mean scales in $\mathcal{O}(dNR)$ time.

In the case of GBQ over a Gaussian measure, if $dRZ < N^2$, GBQ mean-calculation is also $\mathcal{O}(N^3)$ as in traditional BQ. For GBQ over a uniform measure, if $dR < N^2$, we can assume the same.

## 5 EXPERIMENTS

We demonstrate here the empirical results of GBQ compared to traditional Monte Carlo quadrature methods and BQ. Specifically, we measure percent error versus the analytical integral solution, with baselines of Monte Carlo (MC) integration, quasi Monte Carlo (QMC) using Halton sequence sampling (Halton, 1960) over a uniform hypercube, and BQ with the RBF kernel and a Gaussian measure.

For GBQ, we present results in the form of GBQ-Measure-Kernel, where the kernel is chosen from the RFF estimates to the RBF, Matérn 1/2 (M1/2), Matérn 3/2 (M3/2), and Matérn 5/2 (M5/2), and the measure is either uniform (U) or Gaussian (G). We hold static the number of integrand observations $f(\boldsymbol{x})$ available across all baselines and GBQ models. Additionally, we use the same GP kernel hyperpa-

rameters $\boldsymbol{\theta}$ in both BQ and GBQ, which are trained once per each experiment at each $n$ and shared across all models and kernels. For Fourier features $\boldsymbol{\omega}$ and $\boldsymbol{\rho}$ in equations (9) and (11), we sample using Halton sequences as well to produce a smoother coverage of the sample space. Finally, we implement these methods in Julia (Bezanson et al., 2015), and code has been made available [2].

We note that while our experiments consider the employment of the Matérn family of kernels, any shift-invariant kernel can be used in the GBQ integrand prior to adapt to a wide array of problem settings. While there are various analytical solutions to the Matérn family in traditional BQ, they require a kernel-specific integral to be calculated and implemented, and don't exist over all measures $p(\boldsymbol{x})$. We provide evidence to the flexibility of our method by showing that Matérn kernels can be implemented without change of problem formulation by simply sampling features $\boldsymbol{\omega}$ according to the appropriate frequency distribution.

For all experiments, at each training size $n$ we report results as the average model-wise results over multiple runs under different random seeds, and include information on the error variance over runs. While experiments were run for all kernel-measure combinations for GBQ, for brevity we include here only those models that performed best on a given experiment.

## 5.1 1D EXPERIMENTS

Our first experiment is a simple 1D polynomial to empirically verify our theoretical results of section 4 regarding the efficacy of the GBQ method in both recreating results of traditional BQ using the RBF kernel as well as demonstrate the flexibility of kernel choice that GBQ offers.

We model the integral of a polynomial of the form:

$$f(x) = 0.2x^3(x-4)^2 - 3x - 3\,, \qquad (23)$$

in the first case, and disjointed version of the polynomial

$$f(x) = \begin{cases} 0.2x^3(x-4)^2 - 3x - 3\,, & x < 2.5\,, \\ 0.2x^3(x-4)^2 - 3x - 13\,, & x \geq 2.5\,, \end{cases} \quad (24)$$

in the second. The choice of the disjoint polynomial is in order to assess the value of the flexibility of GBQ in enabling varied kernel choice in BQ, and in this case we leverage Matérn kernels, which typically perform better than the RBF on non-smooth data. We use 100 Fourier features in all GBQ models, and run each experiment 10 times under different seeds at each $n$ and report the aggregated mean and 95% confidence bounds in figure 1.

In the first experiment, which represents a smoother polynomial, BQ and GBQ both outperform QMC in accuracy as a

[2]https://github.com/houstonwarren/GBQ.jl

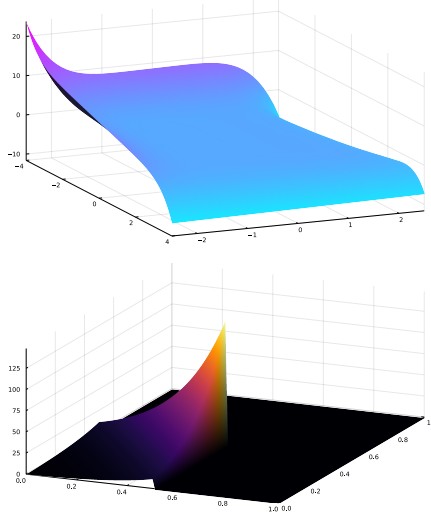

Figure 2: Plots of 2D experiment equations (25) (top) and (26) (bottom).

function of data scarcity. We can see that GBQ-G-RBF is an excellent approximation to BQ, which similarly leverages an RBF kernel over a Gaussian measure, which helps to validate our theoretical results on both the accuracy of the RFF-based integration of the RFF-RBF kernel over a Gaussian measure, as well as the ability for RFFs to parametrize Gaussian distributions.

In the disjoint case, we see that at low $n$, GBQ has a slight advantage over BQ when using the Matérn kernel, but that results converge for all methods as training size increases. While QMC achieves better error at some points, it generally displays more variance over $n$ in this experiment than the BQ and GBQ-based models.

## 5.2 2D EXPERIMENTS

We now move to a selection of 2D experiments, first of which is estimating the integral of a polynomial of the form

$$f(x, y) = -0.005x^4 * 0.1x^3 + y^5(0.02x - 0.08) \\ - 0.001y^2 + 0.2y + 0.5 \quad (25)$$

over the interval $x \in [-4, 4], y \in [-2.5, 2.5]$, as well as a disjoint 2D function:

$$f(x, y) = \begin{cases} e^{5x+5y}, & x < 0.5, y < 0.5\,, \\ 0, & x \geq 2.5, y \geq 2.5\,, \end{cases} \quad (26)$$

over the unit cube.

We perform both experiments over a range of training data sizes from 10 to 1000 $n$, with 5 runs per $n$ at different random seeds. All GBQ models use 300 Fourier features. Plots of these functions can be seen in figure 2, and the

Table 1: 2D Polynomial of Equation (25). Integration Results (% Error).

| $N$ | QMC | BQ | GBQ-U RBF | GBQ-G RBF | GBQ-G M5/2 |
|---|---|---|---|---|---|
| 10 | $98.78 \pm 7.23$ | $8.57 \pm 6.77$ | $17.03 \pm 9.06$ | $10.27 \pm 5.32$ | $\mathbf{4.88 \pm 3.73}$ |
| 25 | $76.57 \pm 16.34$ | $9.69 \pm 7.45$ | $\mathbf{8.32 \pm 7.16}$ | $8.53 \pm 7.39$ | $11.08 \pm 10.63$ |
| 50 | $44.92 \pm 5.7$ | $7.81 \pm 2.64$ | $14.77 \pm 2.6$ | $7.33 \pm 3.07$ | $\mathbf{5.72 \pm 5.22}$ |
| 100 | $31.02 \pm 3.46$ | $4.02 \pm 3.5$ | $\mathbf{1.97 \pm 0.88}$ | $4.04 \pm 2.93$ | $2.41 \pm 1.71$ |
| 250 | $7.97 \pm 1.6$ | $1.22 \pm 1.13$ | $\mathbf{1.03 \pm 0.93}$ | $2.14 \pm 0.77$ | $1.86 \pm 1.1$ |
| 500 | $6.07 \pm 0.85$ | $0.68 \pm 0.63$ | $\mathbf{0.49 \pm 0.53}$ | $1.34 \pm 1.6$ | $1.56 \pm 1.65$ |
| 750 | $5.51 \pm 0.65$ | $0.73 \pm 0.26$ | $\mathbf{0.48 \pm 0.38}$ | $1.22 \pm 1.24$ | $1.35 \pm 1.21$ |
| 1000 | $3.94 \pm 0.46$ | $0.41 \pm 0.26$ | $\mathbf{0.36 \pm 0.26}$ | $1.41 \pm 1.36$ | $1.52 \pm 1.34$ |

Table 2: 2D Disjoint Polynomial of Equation (26). Integration Results (% Error).

| $N$ | QMC | BQ | GBQ-U RBF | GBQ-U M1/2 | GBQ-G RBF |
|---|---|---|---|---|---|
| 10 | $164.04 \pm 0.34$ | $38.42 \pm 0.72$ | $\mathbf{8.26 \pm 3.82}$ | $95.64 \pm 12.34$ | $30.79 \pm 3.68$ |
| 25 | $20.28 \pm 0.75$ | $10.59 \pm 0.75$ | $\mathbf{2.64 \pm 1.0}$ | $5.06 \pm 4.49$ | $10.17 \pm 0.95$ |
| 50 | $23.38 \pm 0.28$ | $26.08 \pm 0.3$ | $17.42 \pm 0.7$ | $\mathbf{12.96 \pm 10.34}$ | $27.14 \pm 0.69$ |
| 100 | $26.8 \pm 0.24$ | $38.93 \pm 0.23$ | $25.06 \pm 0.3$ | $\mathbf{5.92 \pm 7.4}$ | $38.26 \pm 0.28$ |
| 250 | $4.41 \pm 0.16$ | $11.99 \pm 0.16$ | $\mathbf{2.74 \pm 0.33}$ | $2.99 \pm 2.03$ | $12.01 \pm 0.28$ |
| 500 | $3.48 \pm 0.09$ | $12.63 \pm 0.09$ | $3.46 \pm 0.12$ | $\mathbf{2.08 \pm 0.85}$ | $12.68 \pm 0.1$ |
| 750 | $3.24 \pm 0.07$ | $12.38 \pm 0.07$ | $3.01 \pm 0.06$ | $\mathbf{2.02 \pm 0.58}$ | $12.24 \pm 0.1$ |
| 1000 | $0.86 \pm 0.05$ | $9.62 \pm 0.05$ | $\mathbf{0.61 \pm 0.05}$ | $0.73 \pm 0.18$ | $9.48 \pm 0.08$ |

means and standard deviations of the results are reported in tables 1 and 2.

In both experiments, we see that GBQ methods have universally lower mean error than QMC and BQ. The best performing kernel varies across $n$, but in several cases we see that the Matérn has the lowest error, supporting the case that flexibility of kernel choice is a valuable addition to the BQ method when considering both different integrand types as well as available training data.

In the disjoint polynomial experiment, we intentionally include GBQ-G with the RBF (the BQ equivalent) in table 2, even though it was not high performing among the GBQ methods, to demonstrate the potential performance enhancement GBQ offers through kernel choice. We see GBQ-G-RBF track closely with BQ, while GBQ-U with the Matérn 1/2 and GBQ-U-RBF in combination perform better at all $n$, and frequently with implied worst-case error bounds well below the BQ mean error.

### 5.3 5D EXPERIMENTS

We use a 5D problem from a seminal BQ paper (Ghahramani and C. Rasmussen, 2003) to provide an initial evaluation of GBQ in higher dimensions. We model the equation:

$$f(\boldsymbol{x}) = 10\sin(\pi x_1 x_2) + 20(x_3 - 0.5) + 10x_4 + 5x_5, \quad (27)$$

as well as a disjoint variant:

$$f^*(\boldsymbol{x}) = \begin{cases} f(\boldsymbol{x}) & x_i \leq 0.5 \, \forall \, i \,, \\ 4 \times f(\boldsymbol{x}) & x_i > 0.5 \, \forall \, i \,, \end{cases} \quad (28)$$

where observations $y = f(\boldsymbol{x}) + \epsilon$ and $y^* = f^*(\boldsymbol{x}) + \epsilon$ have added noise $\epsilon \sim \mathcal{N}(0, \frac{1}{2})$. We perform integration methods over the 5D unit hypercube using 100 Fourier features. Shortened results are provided in tables 3 and 4 as the average and standard deviation of integral approximation percent error versus the analytical solution across 10 random seeds. Results across all $n$ are available in the supplement.

In the non-disjoint setting, GBQ methods are the highest performing across all experiments with $n > 50$. Notably, we choose to report MC as a baseline other than QMC, as across both experiments in 5D we see a degradation of QMC methods in favor of simple MC. In the disjoint setting, MC is the highest performing at low N, with BQ and GBQ methods performing best at mid to high $n$.

An interesting experimental result was the importance of consistent methodology used for solving the kernel mean $\boldsymbol{\mu_x}(\boldsymbol{X})$ and producing the kernel matrix $\boldsymbol{K}$, when applied in the BQ posterior mean formulation (6). Anecdotally, we found that using the combination of a kernel mean derived from traditional BQ and a kernel that was estimated through RFFs, and vice-versa, produced significantly unstable posterior integral mean estimates. These results suggest the benefit of using the full-stack GBQ method with RFF parametrization of both the kernel and measure distribution in order to achieve the best experimental results.

Table 3: 5D Equation 27 Integration Results (% Error).

| $N$ | MC | BQ | GBQ-U RBF | GBQ-G RBF |
|---|---|---|---|---|
| 10 | **9.67 ± 8.43** | 20.39 ± 3.85 | 23.77 ± 4.33 | 20.35 ± 3.99 |
| 25 | 9.32 ± 7.7 | 3.21 ± 1.87 | 6.02 ± 2.46 | **3.0 ± 1.97** |
| 50 | 5.57 ± 4.14 | **0.61 ± 0.34** | 2.48 ± 0.51 | 0.88 ± 0.42 |
| 100 | 3.81 ± 2.1 | 2.05 ± 0.35 | **0.89 ± 0.44** | 2.25 ± 0.4 |
| 400 | 2.74 ± 1.7 | 2.28 ± 0.2 | **0.33 ± 0.14** | 2.44 ± 0.2 |
| 700 | 2.39 ± 2.43 | 2.29 ± 0.12 | **0.16 ± 0.1** | 2.43 ± 0.16 |
| 1000 | 1.79 ± 1.09 | 2.22 ± 0.08 | **0.14 ± 0.09** | 2.37 ± 0.13 |

Table 4: 5D Equation 28 Integration Results (% Error).

| $N$ | MC | BQ | GBQ-G RBF | GBQ-G M3/2 |
|---|---|---|---|---|
| 10 | **23.94 ± 13.0** | 33.32 ± 3.0 | 33.26 ± 3.12 | 38.11 ± 3.78 |
| 25 | **16.84 ± 20.99** | 18.26 ± 0.86 | 17.96 ± 1.08 | 22.17 ± 1.22 |
| 50 | **7.58 ± 5.92** | 15.15 ± 0.59 | 14.87 ± 0.6 | 16.69 ± 0.7 |
| 100 | 5.89 ± 3.64 | **1.71 ± 0.83** | 2.06 ± 1.28 | 5.53 ± 4.55 |
| 400 | 3.98 ± 2.28 | 1.11 ± 0.42 | 1.7 ± 0.55 | **0.79 ± 0.64** |
| 700 | 3.93 ± 2.51 | 1.03 ± 0.34 | 1.31 ± 0.61 | **0.85 ± 0.44** |
| 1000 | 3.24 ± 2.15 | **0.38 ± 0.24** | 0.89 ± 0.46 | 0.53 ± 0.5 |

# 6  DISCUSSION

In this paper, we have introduced generalized Bayesian quadrature, a method for performing Bayesian quadrature using any shift-invariant kernel while maintaining posterior tractability. We derive the upper bound on the error of this approximation, while also demonstrating the practical benefits on a selection of quadrature problems when compared to traditional numerical integration methods and baseline BQ.

More broadly, we note the wider applicability of the methods proposed in this paper. Our chief theoretical contribution comes within the framework of Bayesian quadrature, but in essence it is providing the analytical solution to a kernel mean when the kernel and measure distribution are approximated by RFFs. However, kernel means have a wide array of use cases as discussed in 2, and represent fertile ground for future applications of our theoretical results.

Additionally, as part of the process of applying GBQ over closed-bounds in multiple dimensions, raising necessity for a truncation term composed of multivariate cumulative distribution functions, we devised a method to parametrize distributions using RFFs and analytically integrate this estimate in order to produce a CDF. For many distributions which offer no closed-form multivariate CDF, this method might be of use.

Future research may look into these applications as well as extending the flexibility and computational aspects of the method. Potential extensions include learning the RFF kernel through its spectral density, leveraging low-rank GP posteriors for computational efficiency improvements in

kernel matrix inversion in the BQ mean, and composing multiple levels of GBQ together into deeper architectures for applications to highly nonlinear problems. The introduction proposed in this paper has demonstrated both theoretical and empirical promise that will provide a solid launching point for these pursuits.

**Acknowledgements**

Rafael Oliveira was supported by the Medical Research Future Fund for Applied Artificial Intelligence in Health Care grant (MRFAI000097).

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
