# OpenReview forum: "Generalized Bayesian Quadrature with Spectral Kernels"
_auai.org/UAI/2022/Conference — UAI 2022 Poster_

### Official Review · Reviewer_ofQc · 2022-04-13

**Q2(1) Originality/Novelty:** 3
**Q2(2) Significance/Impact:** 3
**Q2(3) Correctness/Technical Quality:** 3
**Q2(6) Clarity Of Writing:** 3
**Q6 Overall Score:** 6
**Q8 Confidence In Your Score:** 4

**Q1 Summary And Contributions:**

Bayesian quadrature is an effective approach to computing high-dimensional integrals of a function f with respect to an underlying measure by approximating f through a GP prior.  However, the choice of kernels is usually limited to those kernels for which the kernel means (with respect to the measure) are analytically tractable.  The authors use random Fourier expansions of the kernels (and the measures) to derive BQ for more general kernels.

**Q2 Assessment Of The Paper:**

More detailed information regarding each of these aspects is given below:

**Q2(4) Quality Of Experiments (Optional):**

3: Good: The experimental evaluation is adequate, and the results convincingly support the main claims.

**Q2(5) Reproducibility:**

4: Excellent: Key resources (e.g., proofs, code, data) are available and key details (e.g., proof sketches, experimental setup) are comprehensively described for competent researchers to confidently and easily reproduce the main results.

**Q3 Main Strengths:**

I think the key idea of the paper is really cool: random Fourier feature expansions are a general tool to approximate shift-invariant kernels, and the kernel means needed for Bayesian quadrature can be easily estimated for RFF kernel approximations when the relevant measure (like the kernel) is positive definite and it is easy to sample from the Fourier transformed version of that measure.  I also like that there is some error analysis, though I have not checked all details.

**Q4 Main Weakness:**

Sample efficient high-dimensional integration is intractable without a lot of smoothness or some other special assumption.  But the selling point in the 1D and 2D experiments seems to be that the method can deal with non-smooth integrands.  I am skeptical that this scales to higher dimensions.

**Q5 Detailed Comments To The Authors:**

In low-dimensional spaces, conventional numerical quadrature methods (Gauss quadrature and adaptive quadrature rules) are quite efficient.  In spaces of moderately high dimension, tensor-product quadrature becomes infeasible, though sparse grid methods are sometimes still attractive; but techniques like Bayesian quadrature and quasi-Monte Carlo methods still have a strong edge over the competition.  In truly high-dimensional spaces, both function approximation and quadrature suffer from the curse of dimensionality *unless* the regularity of the function of interest (or sometimes some other measure of function complexity) goes up concurrently with the dimension.  This is part of the great appeal of Bayesian quadrature with a squared exponential radial basis function: implicit in the choice of kernel is the idea that the integrand is smooth enough for this to be a sensible way of doing things.  The proposed technique extends the set of kernels that can be used for modeling the integrand, but does not seem as useful for extending the set of measures that can be treated.  Additional flexibility in the kernel, allowing choice of something less regular, may help with modeling non-smooth integrands in low-dimensional spaces, but it does not seem as likely to help get around the need for a lot of samples in similar situations in high-dimensional spaces.

I could potentially be convinced that I'm wrong about this, but it would require more than 1-D and 2-D test problems (a 5-10 dimensional test problem would be fine to make the point).

It would be helpful to introduce the underlying measure earlier, I think (e.g. writing integral f(x) p(x) dx at the outset, rather than just writing integral f(x) and leaving the measure implicit).

In equation (10), please use either the equal sign or \mapsto, but not both.

I was very confused around (11).  In the statement p(x) is approximately q(x), what is p(x)?  If it is a Gaussian, how is q(x) an approximation?  Also, the normalization constant involves the determinant of a covariance matrix, but the covariance matrix does not appear elsewhere in the expression; is \Sigma = I in this example?

Lemma 1 seems to be about estimating the integrand, not estimating the integral (though the text indicates the latter).

It was a little unclear to me what measures were being used for the standard Bayesian quadrature and QMC methods in the experiment.  I can guess that BQ was with respect to a Gaussian measure and QMC with respect to uniform on a box, but neither of those is clear (the Box transform is as good for quasirandom uniform variates as it is for standard PRNGs).

**Q7 Justification For Your Score:**

I am not fully convinced that this will be much more useful than BQ with squared exponential kernels in practical settings -- sample-efficient high-dimensional integral approximation requires a lot of smoothness (or some other structural assumptions) in a way that seems to undercut the main selling point of using very flexible kernels.  However, I think the key idea is cool, and there are other reasons beyond Bayesian quadrature to want to compute kernel means with different kernels.

**Q9 Complying With Reviewing Instructions:**

1: Yes.

---

### Official Review · Reviewer_zfAV · 2022-04-13

**Q2(1) Originality/Novelty:** 3
**Q2(2) Significance/Impact:** 3
**Q2(3) Correctness/Technical Quality:** 3
**Q2(6) Clarity Of Writing:** 3
**Q6 Overall Score:** 4
**Q8 Confidence In Your Score:** 4

**Q1 Summary And Contributions:**

This paper aims to extend the Bayesian Quadrature (BQ) framework to any shift-invariant by leveraging Fourier random features (RFFs). The idea is to approximate the density function in BQ formulation by RFFs such that the mean estimation of a function over the density measure can have an analytic form. Error bound analysis of the density approximation as well as of the mean estimation are provided. Empirical evaluation of the proposed generalized BQ is presented in both 1-D and 2-D cases.


**Q2 Assessment Of The Paper:**

More detailed information regarding each of these aspects is given below:

**Q2(4) Quality Of Experiments (Optional):**

2: Fair: The experimental evaluation is weak: important baselines are missing, or the results do not adequately support the main claims.

**Q2(5) Reproducibility:**

2: Fair: Key resources (e.g., proofs, code, data) are unavailable but key details (e.g., proof sketches, experimental setup) are sufficiently well-described for an expert to confidently reproduce the main results.

**Q3 Main Strengths:**

- Novelty: The proposed generalized BQ discovers a new class of tractable BQ problems when both kernels, as well as density functions, have their representations as RFFs, and it allows the BQ to be applied with more expressive kernels.
- Writing: This paper is overall well-written. It has provided enough background to understand the proposed methods and it is easy to follow.

**Q4 Main Weakness:**

Discussion on the runtime complexity is missing in this work while the complexity of the proposed algorithm might be the main issue. From equation (13), it seems that the computation of the mean estimation is quadratic in the number of Fourier random features. Moreover, in the empirical evaluation, the number of features are 100 and 300 in the 1D and 2D experiments respectively, meaning that it requires a non-trivial number of features to get decent results. I think the authors should also include the comparison of runtime in their experiments for a fair comparison and to better illustrate the efficiency of the proposed approach. Also, only examples in 1D and 2D are presented, while the previous BQ paper such as [1] has their algorithm run on cases with dimensions being 5 or 15 where the scalability of the algorithms would be better illustrated than the low dimensional cases.
[1] Kandasamy, Kirthevasan et al. “Bayesian active learning for posterior estimation.” AAAI 15

**Q5 Detailed Comments To The Authors:**

- Can the authors confirm what is the runtime complexity of generalized BQ and whether scalability is an issue?

- In Sec 5.2, it mentions that the size of training data n ranges from 10 to 1000, while it is unclear what values does n exactly take. It is unclear for which n it gives the results in Table 1 & 2 and also it seems that not all the results for n are presented.

- Suggestion on Definition 1: I think the authors should put only the definition of generalized BQ for any density, and separate the cases when the measures are Gaussian or Uniform as Propositions/Theorems instead of including everything in the definition.

- Typo: Before Eq (11), the limit should be 'R -> \infty' instead of 'r -> \infty'.


**Q7 Justification For Your Score:**

My main concern is the scalability of the proposed generalized BQ. Also, the empirical evaluation presented in this paper is on a very light side and not well presented.

**Q9 Complying With Reviewing Instructions:**

1: Yes.

---

### Official Review · Reviewer_JPgc · 2022-04-14

**Q2(1) Originality/Novelty:** 3
**Q2(2) Significance/Impact:** 3
**Q2(3) Correctness/Technical Quality:** 3
**Q2(6) Clarity Of Writing:** 4
**Q6 Overall Score:** 7
**Q8 Confidence In Your Score:** 3

**Q1 Summary And Contributions:**

The authors propose generalised Bayesian quadrature (GBQ), a form of Bayesian quadrature that enables using any shift-invariant kernel in the integrand GP model while maintaining the analytical tractability of the integral posterior. The authors use the spectral representation of shift invariant kernels and their random Fourier features. They do so by showing that a Gaussian density can be approximated by RFFs. The authors compare GBQ to existing numerical integration methods and standard BQ.

**Q2 Assessment Of The Paper:**

More detailed information regarding each of these aspects is given below:

**Q2(5) Reproducibility:**

4: Excellent: Key resources (e.g., proofs, code, data) are available and key details (e.g., proof sketches, experimental setup) are comprehensively described for competent researchers to confidently and easily reproduce the main results.

**Q3 Main Strengths:**

* Very well written
*  The authors use both the uniform and the Gaussian measures in the experiments section. The Gaussian case can be compared directly with BQ that uses RBF kernels. The uniform measure works well, there is no competing method.

**Q4 Main Weakness:**

- It would be good to have some comparison of computational time. Mention if there are any additional steps that increase the computational cost of GBQ wrt BQ. Also, in terms of the dimensionality of the input space. Please clarify and ideally add more through experiments wrt dimensionality for a high enough dimension.
- Edit: the authors have provided a computational complexity analysis in the rebuttal. I encourage the authors to include the  in the paper since it gives a full picture of this.

**Q5 Detailed Comments To The Authors:**

- I suggest to change the colour scheme in Figure 1, red and orange can be easily confused
- In the text, the authors say that the Matern kernel should work well with the disjoint polynomial but then in the figure you can see it has the highest error in the small sample size regime, please clarify


**Q7 Justification For Your Score:**

- The paper is well structured and clearly written. The authors present a technically sound contribution that extends the usability of Bayesian quadrature. I would like to see more fleshing out of the computational cost and how the approximation error increases as well as what happens in a high dimensional space.

I’ve read the other reviewers feedback as well as the authors replies. The authors have addressed my concerns so I am increasing my score accordingly.

**Q9 Complying With Reviewing Instructions:**

1: Yes.

---

### Official Review · Reviewer_2iQF · 2022-04-16

**Q2(1) Originality/Novelty:** 3
**Q2(2) Significance/Impact:** 3
**Q2(3) Correctness/Technical Quality:** 4
**Q2(6) Clarity Of Writing:** 4
**Q6 Overall Score:** 8
**Q8 Confidence In Your Score:** 5

**Q1 Summary And Contributions:**

This work is about the computation of integrals via Gaussian processes (GP), also known as Bayesian quadrature (BQ). The authors propose a generalized version of Bayesian quadrature by exploiting random Fourier features twice---once for the kernel and once for the measure in the GP's posterior mean formulation.

**Q2 Assessment Of The Paper:**

More detailed information regarding each of these aspects is given below:

**Q2(4) Quality Of Experiments (Optional):**

3: Good: The experimental evaluation is adequate, and the results convincingly support the main claims.

**Q2(5) Reproducibility:**

4: Excellent: Key resources (e.g., proofs, code, data) are available and key details (e.g., proof sketches, experimental setup) are comprehensively described for competent researchers to confidently and easily reproduce the main results.

**Q3 Main Strengths:**

+ The practical approximation of integrals is one of the most important practical mathematical problems. Only a handful of viable methods exists for that problem and the authors contribute a true improvement over the state-of-the-art.
+ The paper is well written an a pleasure to read. The formal presentation is very good and it is easy to follow the proofs and the author's reasoning.
+ An implementation is available.

**Q4 Main Weakness:**

- The limitations are not discussed. Is the applicability limited, e.g., by assumptions on k(0,0) or the positive definiteness of "p"? What about the methods complexity? Is it more computationally demanding than standard BQ?
- The experiments have rather small scale. I would love to see results on non-toy data. What are the computational limits of this method?
- Non-BQ stochastic quadrature methods are not discussed / mentioned.

**Q5 Detailed Comments To The Authors:**

Honestly, I am a fan of this work. The idea is brilliant and it even works in practice.

The proposed method works with any shift-invariant kernel, which simplifies the usage in practice when compared to standard BQ. After explaining nicely the intuition behind the approach, the authors present an upper bound on the quadrature error followed by a careful empirical evaluation of their approach. On small-scale benchmark data, traditional Monte Carlo techniques as well as standard BQ deliver far worse results than the proposed method, especially when data is scarce.

One question: In Eq.(11), it is unclear why we see Sigma in the normalizing constant when exp(-|x-mu|^2) has variance = 1?

**Q7 Justification For Your Score:**

For me, the most important point is the fact that the authors contribute a theoretically sound method for one of the hardest and most versatile practical mathematical problems of AI.

My score can only be improved by experimental results on large scale problems.

**Q9 Complying With Reviewing Instructions:**

1: Yes.

---

### Decision · Program_Chairs · 2022-05-15

**Decision:**

Accept (Poster)

**Comment:**

Meta Review: While there are mixed reviews for the paper, concerns largely relate to computational complexity which seem well addressed by the authors in their responses. The broader coverage of the literature in the paper is not great. For example, it's surprising that very well known work about spectral kernels, such as spectral mixture kernels (http://proceedings.mlr.press/v28/wilson13.pdf), is discussed nowhere in the text. This should definitely be corrected in a final version, as well as a careful accounting of reviewer concerns.